## POINT OF VIEW

# Motivating participation in open science by examining researcher incentives

**Abstract** Support for open science is growing, but motivating researchers to participate in open science can be challenging. This in-depth qualitative study draws on interviews with researchers and staff at the Montreal Neurological Institute and Hospital during the development of its open science policy. Using thematic content analysis, we explore attitudes toward open science, the motivations and disincentives to participate, the role of patients, and attitudes to the eschewal of intellectual property rights. To be successful, an open science policy must clearly lay out expectations, boundaries and mechanisms by which researchers can engage, and must be shaped to explicitly support their values and those of key partners, including patients, research participants and industry collaborators.
DOI: https://doi.org/10.7554/eLife.29319.001

SARAH E ALI-KHAN, LIAM W HARRIS AND E RICHARD GOLD*

*For correspondence: richard.
gold2@mcgill.ca

## Introduction

Open science involves making scientific knowledge (in the form of papers, data, research tools, and other outputs) openly and practically available in order to accelerate discovery, innovation and clinical translation (*Gold, 2016*; *Global Alliance for Genomics and Health, 2016*; *Fecher and Friesike, 2014*). In a world first for an academic research institution, in 2016 the Montreal Neurological Institute and Hospital (the MNI) adopted a broad open science policy across all of its research activities. The MNI committed to make publicly available all positive and negative data by the date of first publication, to open its biobank to registered researchers and, perhaps most significantly, to withdraw its support of patenting on any direct research outputs (*Figure 1*; *Poupon et al., 2017*; *Owens, 2016a*; *Rouleau, 2017*). The span of this openness raises unique hopes and concerns for stakeholders, and also offers an important opportunity to examine and understand the potential benefits and costs of open science.

Despite the benefits offered by open science, it has proven difficult to implement its

ideals (*Edwards, 2016*; *Nelson, 2009*; *Open Research Data Task Force, 2017*). Recent evidence suggests that providing incentives for individual researchers to participate is the key rate-limiting step (*Kidwell et al., 2016*; *Longo and Drazen, 2016*; *Fecher et al., 2015*). This has stimulated many to think more deeply about how cultural factors, policies and metrics may affect the uptake of open practices by researchers (*European Commission, 2017*; *Wilsdon, 2015*; *Harley, 2013*; *Neylon, 2017*). This study builds on this work by providing insight into the perspectives of researchers in the lead up to the adoption of an open science policy within their institution. We conducted semi-structured interviews with 25 researchers, staff and partners, approximately one-third of the MNI's constituency, to identify their attitudes toward open science, the incentives they felt necessary to engage in open science, disincentives, the role of patients, and the eschewal of intellectual property rights. We then applied qualitative thematic content analysis to these data to identify key topics for discussion and

**Figure 1.** Guiding principles for the conduct of open science at the Montreal Neurological Institute and Hospital (MNI). These principles cover five areas: the public release of data and other scientific resources; external research partnerships; the MNI Biobank; researcher and patient autonomy; and intellectual property. The authors developed draft Guiding Principles based on the results of this study. This draft was then presented to the MNI staff, management and researchers, who reviewed and amended the draft during two rounds of discussion and feedback. These Guiding Principles were adopted by the MNI in December 2016.

DOI: https://doi.org/10.7554/eLife.29319.002

to derive the framework for the MNI's Open Science Policy (*Figure 1*).

## Results

Our analysis yielded a number of key themes which we summarize below. Please see *Ali-Khan et al. (2015)* for the full research report, and *MNI, 2017* for the latest information about the initiative. In this manuscript, we sometimes use proportions or numbers to quantify the number of interviewees who expressed a particular opinion: please see 'Materials and methods: Reporting' below for further details and a caveat.

### Nebulous definition breeds uncertainty

Our analysis revealed that MNI researchers were generally favorable toward the notion of open science, reporting that they already engage in significant sharing. Nevertheless, almost half our interviewees noted that the concept is vague (*Fecher and Friesike, 2014*; *Grubb and*

*Easterbrook, 2011*; *OECD, 2015*) and expressed uncertainty about what precisely open science would require of them (*Open Research Data Task Force, 2017*; *Neylon, 2017*; *Ferguson, 2014*). Based on our data, this uncertainty can discourage participation in two ways. First, the absence of clearly articulated definitions raised fears that open science practice may extend beyond comfort levels. Conversely, those who desired greater openness were cautious about taking part lest the lack of clarity prevent meaningful change.

### Open science needs to be stakeholder driven

In the context of the MNI's initiative, just over half of researchers stated that an institution-level open science policy could only be successful if it was developed in close collaboration with researchers, rather than simply imposed in a top-down fashion by management. Researchers were concerned that a broad one-size-fits-all directive may have adverse effects on their scientific priority and curtail their academic freedom by denying them, for example, the right to choose where to publish (*Levin et al., 2016*), to patent findings (*Murray, 2010*), to publish first and to control access to research outputs (*Fecher et al., 2015*).

A further dominant opinion, explicitly mentioned by over a quarter of interviewees, was that open science should not be implemented for 'its own sake'. Instead, policy should reflect the values and needs of key stakeholders. Misgivings about open science were always tied to perceived negative effects on their careers, on their relationships with the patients who participate in research projects, or on their collaborations with industry.

### Current state of sharing

More than three quarters of interviewees reported that they already share data and research tools on request after publication as a standard practice, and that they share with established collaborators before publication. However, outside of the bioinformatics and imaging area, few engaged in large-scale public sharing. Willingness to share varied depending on the data type and the resources involved in their development. Unsurprisingly, interviewees noted that digital data is more shareable. Most said that patients' clinical and genomic data should be widely shared, subject to consent and ethics protections (*Kaye, 2012*), due to the

limited intellectual contribution made by the researcher who collected the data. Four researchers felt that these data belonged to patients and thus should not be hoarded. On the other hand, two interviewees said it would be unfair to force early sharing of data and materials yielded through resource-intensive processes, such as iPS cell lines, or rare and depletable biosamples. Overall, there was wide agreement among interviewees that it is appropriate to wait until after publication to share experimental data beyond established collaborators. This was true even in the case of bioinformatics and imaging researchers, whom we expected to be the most comfortable with open pre-publication sharing.

### Researchers are motivated by ethics and also by career advancement

Approximately three quarters of our interviewees articulated ethical motivations for more rapid, open sharing of data and scientific resources. These include the belief that publicly-funded research outputs ought to be released with minimal delay and that consistent with patients' wishes, researchers have a duty to ensure that samples are broadly shared to maximize research and discovery. Likewise, many alluded to the negative impact of data 'bottlenecking' before publication, and the increased efficiency that would result from earlier and greater access to research outputs (*Fecher et al., 2015*).

Career advancement was an even more prevalent motivation. All but one interviewee mentioned the potential for open science to bolster their professional standing. Many said that requests for reagents, data or other tools led to diverse and unanticipated collaborations, expanding their interests, visibility and professional impact. Others noted that sharing has a positive impact on publication productivity and citations (*Piwowar and Vision, 2013*). While many who currently share stated that this is good academic practice, interviewees said they did so anticipating reciprocal behavior (*Edwards, 2016*).

Related to this, almost all interviewees raised a fundamental concern: that open science may put them at a professional disadvantage if they were compelled to release data and other resources before they have extracted value from their work (*Fecher et al., 2015*; *Harley, 2010*; *Levin et al., 2016*; *LERU Research Data Working Group, 2013*). Over three-quarters said that they would like to share more broadly and

earlier in the research process, but they feared being 'scooped' and/or compromising the possibility of publishing in a high-impact journal. Further, a few were concerned that the MNI would go too far 'out on a limb' by adopting open science before other institutions in Montreal, Canada and beyond followed suit. In this case, others might benefit from MNI resources without reciprocating.

Most interviewees stated a preference to fully understand and publish before publicly releasing a dataset or reagent. Further, approximately half of researchers mentioned that it is socially irresponsible to release data before it is fully validated and its quality is assured, as this could waste others' time and promote erroneous conclusions. Several noted the potential for shared data to be used in research or for applications with which they do not agree (*Wouters and Haak, 2017*). They worried that their or the MNI's reputation might be compromised by such associations. Thus, while nearly all interviewees said the culture of science is shifting toward greater openness, more than half cautioned that the MNI should advance with care.

### Attribution and publication
Almost every interviewee emphasized the highly competitive nature of biomedical research (*Harley, 2010*), and the central importance of maintaining strong academic metrics. A third of interviewees underlined that ensuring proper attribution and returns for sharing would motivate their participation in open science practice (*Wouters and Haak, 2017*). They noted that this could include citation, acknowledgement and in some cases co-authorship for use of shared data, as well as the development of new metrics to measure sharing contributions (so that they could be included in funding and academic advancement decisions).

### Infrastructure and resources
All but one interviewee stressed that openness is time- and resource-intensive, including for example, the need for payment of open access publication fees, and for the preparation, formatting and handling of data and other research outputs for sharing (*Levin et al., 2016*; *LERU Research Data Working Group, 2013*). More broadly, researchers spoke of the costs inherent in the set-up and management of sharing infrastructure, including cyber and biobanking frameworks. Thus, interviewees emphasized that institutional support will be critical to

support open science practice amongst researchers (*Das et al., 2016*; *Poupon et al., 2017*).

### On patients
The majority of interviewees underlined public and patient benefit as the preeminent goal of open science at the MNI (*Rouleau, 2017*). The institution's dual research and clinical functions offer the potential for ongoing data collection from consented patients across multiple research modalities. Opening this platform to outside researchers promises a powerful collaborative discovery resource, but raises familiar concerns about the protection of patients and research participants (*Kaye, 2012*).

The setting in which patients would be engaged about open science research and the nature of informed consent mechanisms were key points of disagreement. Some interviewees favored engagement solely by treating physicians, while other advocated for a centralized location with dedicated staff. Likewise, some supported broad consent to maximize the future use of samples and data, while others said tiered or dynamic consent (*Kaye, 2012*) would ensure that the preferences of research participant were properly addressed. However, almost every interviewee emphasized the primacy of safeguarding both patient confidentiality and the autonomy to decline participation without prejudice.

A few researchers said that patients may be concerned about the potential for their data or samples to be shared with as-yet unknown researchers. However, most thought that patients would be strong advocates for open science, given the framework's promise for advancing new understanding and treatments. Some further suggested that open science would increase patient and public trust (*Royal Society, 2012*; *Grand et al., 2012*) given its emphasis on minimizing intellectual property, increasing transparency and channeling benefits to patients, rather than to industry. They noted this could increase the stature of the MNI in attracting high-quality researchers and trainees, greater patient participation in research and augmented private donation. Finally, several interviewees pointed to the potential for open science to allow patients to become more equal and informed research partners due to greater transparency through the research process. Improvement of researcher-participant relationships could enable more patient-responsive

studies, enhancing patient satisfaction and research outcomes.

### Research ethics

Many interviewees emphasized the key role of the institutional Research Ethics Board (REB) in facilitating their open science practice. One third of interviewees suggested that the REB is sometimes overly stringent and acts as a barrier to efficient sharing of their own research data, or to making use of patient data shared through open web repositories. Conversely, just under two thirds of interviewees said that the REB is evolving in step with open approaches to science, for example partnering with researchers to develop detailed governance mechanisms and patient consent protocols to allow broad sharing (*Das et al., 2016*; *Poupon et al., 2017*). Some interviewees underlined that harmonizing ethical requirements across Canada and internationally is a key priority to realize the promise of global open science. One way that harmonization could accelerate open science would be the introduction of standard click-wrap agreements that are signed when patient-derived data are downloaded, thus avoiding the need use traditional (and cumbersome) side agreements that are signed (and sometimes notarized) by central administrators (*NKI-RS, 2017*).

### On intellectual property, industry and collaboration

Forgoing intellectual property, particularly patents, within the context of the MNI's research activities is unprecedented for a major academic research institution (*Stilgoe, 2016*). Thus, unsurprisingly, the topic of intellectual property polarized opinion. Many of the basic science researchers we interviewed felt that pursuing patents is inefficient, noting that patenting is rarely worth its costs from either a commercial or translational standpoint. Conversely, many clinical-translational researchers said that the freedom to pursue intellectual property is an important aspect of their academic freedom and professional impact. Several underlined the importance of patents in facilitating partnerships with industry stakeholders (*Murray and Stern, 2006*). Only five of this latter group mentioned personal material benefit as a motivation. Further, some conceded that patents are often pursued too early in the research process, which may slow scientific progress. However, despite many of the interviewees not being interested in holding intellectual property themselves, there

was a widespread belief that patents or other modes of legal protection are essential to ensure the translation of research later in the process.

A key area of resistance to open science concerned relationships with industry. These partnerships can bring researchers valuable professional opportunities and offer important avenues for translation and patient benefit. Some interviewees worried that mandated data sharing may deter industry from sponsoring clinical trials. Indeed, the MNI has said it will not impose a specific timeline for release of trial data by sponsors (*Poupon et al., 2017*). However, researchers note that all patient-participant data will be captured within the MNI Open Science Clinical Biological Imaging and Genetic (C-BIG) biobank: this means that when information on a therapeutic molecule is disclosed by collaborators, this information can be easily integrated with the corresponding patient data. Further, some interviewees said that making data and biosamples openly accessible might reduce the attractiveness of MNI researchers to potential collaborators, who may want to operate under a more closed model. However, the public launch of open science at the MNI seems to have stoked interest of companies in a range of sectors including pharmaceuticals, imaging, deep learning, big data and information technology (*Poupon et al., 2017*). The initiative has also impelled major gifts from private donors (*Stilgoe, 2016*).

Interviewees reported that the absence of intellectual property rights by the MNI over discoveries derived from C-BIG biobank materials or data is a key draw for partners. Nevertheless, there are dissenters. Some interviewees and some of the MNI's potential industry collaborators stated that they will proceed with care, 'testing the waters' as they engage in open practices. Others, however, including the multinational biotechnology company Thermo Fisher Scientific, have rapidly secured open-science-based partnerships with the MNI, noting a shared objective in improving the accountability and efficiency of research. Thermo Fisher stated that the MNI's open science policy will greatly accelerate the development and commercialization of more relevant reagents, while allowing the sharing of the experimental data with the research community. However, they underlined that maximizing the benefits of open science depends on broad community adoption.

One key problem highlighted by industry interviewees and MNI management is the legal implications of sharing materials that may be

encumbered by existing intellectual property rights on tools that have been used in their creation (*Rouleau, 2017*). This real, but not insurmountable hurdle, has already slowed negotiations with Thermo Fisher. As Guy Rouleau, director of the MNI, said: "The irony is that this is a perfect example of what open science at the MNI seeks to avoid: intellectual property rights over scientific tools that chill research progress, with high social, economic and opportunity costs."

## Discussion

To date, open science has largely depended on bottom-up forces to drive adoption as funders, publishers and governments have often been reluctant to enforce open science norms (*Open Research Data Task Force, 2017*; *Royal Society, 2012*). Our analysis reinforces the point that researchers' sharing behavior is primarily motivated by rational self-interest rather than pure altruism (*Fecher et al., 2015*; *Levin et al., 2016*; *Haeussler, 2011*). Widespread uptake in competitive research environments requires that open science is not only theoretically attractive but that it is beneficial to stakeholder communities. Our analysis revealed key concerns arising from uncertainty about what open science will require of researchers, and the risks open science may pose for professional competitiveness and for relationships with important research partners. To realize the promise of open science for discovery, innovation, research impact and reproducibility, an open science policy must clearly lay out expectations, boundaries and mechanisms for participation, and must be shaped to explicitly support researchers' values and those of other key stakeholders.

Recent work points to the pivotal role of institutions in influencing open science practice in their constituencies (*Fecher et al., 2015*; *LERU Research Data Working Group, 2013*; *Huang et al., 2012*). In this context, the MNI is an interesting case study because it is the first organization to introduce an institution-wide open-science platform (*Rouleau, 2017*) and because certain characteristics of biomedical research (it is highly competitive, it involves human subjects and the commercial returns can be very high) make it a challenging field for openness (*Tenopir et al., 2011*; *Haeussler, 2011*; *Walsh et al., 2007*). Our study underlines several relevant points.

First, our analysis emphasized the importance of fostering buy-in and trust by developing the MNI open science policy from the ground-up through researcher engagement. Others have also underlined the benefits of this approach (*Open Research Data Task Force, 2017*). This current research, which informed the MNI's framework, represents part of that process. Over the next five years, studies engaging the range of important stakeholders including researchers, patients, philanthropy, governments, research participants and industry partners will contribute to further refining MNI policy (*Gold, 2016*). It has been reported that social milieu can help overcome the reluctance to share (*Haeussler, 2011*). The MNI's policy, sanctioned by all MNI researchers (*Poupon et al., 2017*), may enhance participation compared with situations in which researchers act without specific institutional policy, support or advocacy on their behalf. Moreover, given the emphasis on open science within the MNI, participation may raise researchers' social capital, further encouraging a willingness to share (*Haeussler, 2011*). Increasing adoption of open science practices may place greater pressure on competitors to adopt the same normative practices, as this affects their status in the community (*Westphal et al., 1997*). Thus, MNI policy may begin to shift norms not only within the institute, but across the broader community (*Fauchart and von Hippel, 2008*). We will examine these hypotheses in future work.

Second, our study reiterated the importance of clear policies over ownership and control of scientific outputs. For example, researchers want to benefit from their work before broad release (*Fecher et al., 2015*; *Wouters and Haak, 2017*). Allowing researchers to exercise reasonable choice over when and how they share, and providing the leeway to adapt practice to different research contexts, may reinforce their trust in open science and promote participation. Consistent with this notion, the MNI policy calls for data release before or on the date of first publication and underlines the principle of autonomy (*Figure 1*). Notably, while other studies underline researchers' belief in ownership over data they produced (*Fecher et al., 2015*; *Wouters and Haak, 2017*), some of our interviewees suggested that publicly-funded research data and biosamples belong to the public or to patients, rather than the researchers themselves. This perspective may derive from interviewees' proximity to patients and the needs that open neuroscience seeks to address, or it may be

related to the local culture at the MNI (*Rouleau, 2017*), the local socio-political context or to individual personality differences.

Third, our study, underlined the need for institutional support to encourage open science practice. The MNI is addressing researcher concerns through substantial institutional investment. The development of specialized patient consent and other REB processes, cyber and biobank infrastructure, support staff and streamlined workflow promote clear and efficient sharing (*Das et al., 2016*; *Poupon et al., 2017*). These platforms aim to support researcher trust by enabling the discretion to choose when and what to publically share and by protecting researchers' and partners' interests within a secure sharing environment. Likewise, the MNI recently partnered with F1000 to launch an open research platform (https://mniopenresearch.org) to allow rapid sharing of articles, data and other outputs. The MNI also offers a stipend to support the use of the platform and for open access publication in journals of the researcher's choice. These resources are designed to reduce barriers to participation, acting as carrots rather than sticks to encourage open science practices (*Ferriera, 2008*; *Leonelli et al., 2015*). Tracking the impact of explicit institutional policy and resources compared to their absence will yield important information on the conditions needed to enhance open practices.

Fourth, our analysis corroborates the need to ensure professional returns to promote open science *Fecher et al., 2015*; *LERU Research Data Working Group, 2013*). Developing relevant and effective performance indicators and incorporating them in career advancement and funding processes was highlighted in our study and others (*Peekhaus and Proferes, 2016*; *Borgman, 2015*; *European Commission, 2017*; *Wilsdon, 2015*; *Harley, 2013*). A report on research data in the UK recently noted that professional incentives for researchers to share remain weak at best (*Open Research Data Task Force, 2017*). In the context of open access publication, several studies note that current entrenched academic reward structures can stymie new modes of research and communication (*Peekhaus and Proferes, 2016*; *Eger et al., 2015*; *Xia, 2010*; *Bjork, 2004*). Advocacy, partnership-building and policy-alignment within and across institutions wishing to encourage openness (including governments, funders, journals and publishers) will be important to evolve reward structures that match policy goals and instigate the desired

cultural shift (*Leonelli et al., 2015*; *Neylon and Wu, 2009*; *Munafò et al., 2017*).

Fifth, the role of intellectual property was contentious in our interviews, as has been previously noted (*Ferguson, 2014*). Recent decades have seen a decided policy focus in North America and beyond on encouraging academics to patent with the goal of augmenting translation and university funding (*Nicol, 2008*). However, there is little consistent evidence to support this model's effectiveness (*Kenney and Patton, 2009*; *Williams, 2017*). Open science at the institutional level offers the potential for faster dissemination of academic outputs, greater leverage of research investments, and an emphasis on universities as hubs for knowledge generation and dissemination (*Nicol, 2008*). At the same time, open science is expected to promote the local innovation ecosystem by spreading socio-economic impact (*Gold, 2016*). Thus, the seeming conflict between commercialization and open science agendas may be more apparent than real. Exploration at the interface of these two streams is needed to clarify how they may cross-pollinate to maximize equity, justice and efficiency in exacting research benefits.

## Conclusion

Our study reiterates previously reported researcher concerns toward the adoption of open science practices, and uncovers fresh nuance. In particular, the institutional setting, the broad scope of research undertaken at the MNI and the proximity of patient and industry partners provide a new context to explore these issues. Here, we provide baseline findings that will be important to follow. Ultimately, our analysis underlines that open science should not impose extra burdens on researchers. Rather, policy must make sharing simple and should be structured to enhance researcher competitiveness, research programs and partnerships with patients and industry. Over the next five years, we will monitor the impact of the MNI's open science policy through the collection of scientific, innovation, economic, social, and stakeholder metrics that will be made publicly available (*Gold, 2016*). As quantitative and qualitative data accrue through this large-scale experiment, these data will be fed-back to fine-tune best practices at the MNI. At present, many actors may be cautious, yet as the initiative reaches critical mass, it may begin to generate clear evidence of benefits, shift community norms and expectations, and encourage

participation. The proof of this open science pudding will be in the eating.

## Materials and methods

### Study design

This study uses qualitative research methods: in-depth semi-structured interviews, followed by thematic content analysis. This methodology is best suited to in-depth exploration of the experiences and perceptions of research participants, and the meanings they attach to these (*Braun and Clarke, 2006*; *Patton, 2002*). As such, it is well-aligned with the needs of case studies undertaken to inform policy, as was the current research (*Ritchie and Spencer, 2002*). We report these methods according to Consolidated Criteria for Reporting Qualitative Research (COREQ) guidelines (*Tong et al., 2007*).

### Study sample and research design

Due to the small community of relevant research participants, we used a purposive and snowball sample strategy to select invitees. The interviews were conducted in two phases. The first focused on MNI researchers and staff over Summer 2015. We conducted a second phase of interviews in Fall 2016, to address knowledge gaps concerning the viewpoints of researchers' industry collaborators, and to update perspectives subsequent to launch of the open science initiative (*Owens, 2016b*). Thus, we re-interviewed three researchers and one whom we had not previously engaged, who all interact with industry partners. We also interviewed one industry collaborator, one industry veteran and one not-for-profit collaborator, and obtained a public statement from Thermo Fisher Scientific on its partnership with the MNI.

To begin the first phase of interviews, the MNI management provided a list of research areas and principal investigators that we independently used to selectively invite participants via email. During interviews we asked for suggestions for others who may provide insight, whom we subsequently invited. In the second phase we re-contacted select participants.

### Data collection

One of us (SEA) with LH and/or a colleague (Kendra Levasseur; KL) conducted the interviews. None of these researchers was known to any of the participants prior to receiving the study invitation. ERG, who did not participate in the interviews, may have been known to some, as all are employed at the same institution. We conducted all interviews with MNI researchers face-to-face at locations of the interviewees' choosing, namely their offices or labs, in June through September 2015. Before beginning interviews, we described our credentials, professional positions, the study context, confidentiality and privacy considerations, and noted the independent, academic nature of our research. All interviews with industry or non-profit collaborators were conducted over the phone.

All but one interview was digitally recorded and transcribed verbatim by LH, KL or other research assistants. One participant declined recording, so manual notes were taken. Interviews lasted between 30 and 90 minutes. We developed the interview guide based on our review of the open science literature and knowledge of the MNI's research focus (see http://paceomics.org/index.php/interview-guide/), incorporating the feedback of the MNI management on a first draft. The interview process was iterative, feeding forward key issues raised by interviewees to maximize the informativenes of our research.

To start each interview, we collected demographic information to facilitate comparative analysis across interviews. We then asked open-ended, semi-structured questions to explore individual experience and knowledge. In qualitative research, the dataset is considered complete when a point of theoretical saturation is achieved—meaning no new major ideas, information or themes are emerging from interviews (*Morse, 1995*). Participant recruitment continued until we reached this point, and with consideration for achieving balanced representation across gender and research area. In total, we conducted 21 interviews with MNI researchers and staff in the first phase. In the second we interviewed four MNI researchers, and three external participants as described in 'Study sample', resulting in 25 interviewees in total.

### Qualitative thematic analysis

We analyzed these data using thematic content analysis methods (*Braun and Clarke, 2006*; *Ritchie and Spencer, 2002*) This process consists of seven iterative stages: (i) familiarization; (ii) generation of an initial coding framework and application of these codes to the dataset; (iii) searching for and verification of themes across the entire dataset; (iv) identification of relationships and distinct differences between codes/subgroups of ideas; (v) definition and naming of

themes; (vi) re-reading of the interviews and modifying codes based on emerging themes; and finally (vii) mapping and interpretation of the overall narrative identified from the data. We used NVivo 10 software to organize and manage our analysis (*QSR International, 2016*). This software enables tracking of the research process, thereby facilitating auditability and reproducibility, and thus the credibility of the work.

### Development of analytical categories

During the thematic analysis process we developed several analytic categories. These capture what we defined as 1) 'substantive'; 2)' auxiliary'; and 3)' key' data or issues. 'Substantive' categories captured interviewees' opinions, concerns, ideas and motivations regarding open science. Those categorized as 'auxiliary' captured contextual information that we used to further our understanding of the 'substantive categories', such as the types of resources or stakeholders the interviewees mentioned as relevant. For example, where a researcher expressed concern about sharing iPS cells because of the large investment required to generate them, the text would be coded into a substantive category ("disincentives to sharing or open science – researcher time and money invested in resource creation") and an auxiliary category ("types of resources – iPS cells and other cell lines") to give contextual information. 'Key categories' represent the subset of interviewees' opinions, concerns and motivations that we determined are the most relevant to the development of an open science policy at the MNI. Often, this material represents the most significant sources of disagreement or tension about the proposed shift to open science, and material that was the most emphasized by interviewees. See the full coding framework in the supplementary materials.

### Validity and reliability of analysis

To support the validity of our analysis three coders (SEA, KL and LH) worked together to develop the initial coding framework and ensure it was consistently applied across interview transcripts. This coding framework was informed by our knowledge of the literature and the focus of our research questions (a top-down or deductive approach), in addition to new ideas that we identified inductively from the 'grounded' data (a bottom-up approach; *Patton, 2002*; *Ritchie and Spencer, 2002*). LH and KL then divided the interviews amongst them to code individually. SEA, KL and LH met regularly throughout the research process to compare and discuss data interpretation and modify the framework. The entire research team met on a weekly basis to discuss emerging findings (SEA, LH, KL and ERG).

KL and LH also undertook a formal inter-coder reliability analysis using NVivo 10 software (*QSR International, 2016*), by both coding the same interview and measuring agreement between the two copies. Most inter-coder percentage agreement at each category varied from 80 to 100%, revealing only minimal disagreement between coders. Categories at which agreement was lower were among the 'auxiliary' group. When we inspected discrepancies, we observed that this was due to minor differences in the length of the text coders had selected for auxiliary categories, with minimal implications for our analyses. Finally, we triangulated data across interviews and available policy, academic and web resources, further seeking to ensure the accuracy of our findings.

### Reporting

Our analysis yielded ten major themes of discussion, which we report here as nine themes after combining the intellectual property and collaboration themes to improve the flow of the text (see *Ali-Khan et al., 2015* for the full report). We note that in this manuscript we may indicate the proportion or number of our interviewees who held the described opinions. Two limitations to this approach should be considered. First, our sample, while broadly representative of the units, demographics and responsibilities at the MNI, is not random. Further, we note that in the course of semi-structured interviews, respondents will answer according to their specific interests and knowledge (*Patton, 2002*; *Ritchie and Spencer, 2002*). Therefore, the content of individual interviews is not always directly comparable. Subject to this caution, we believe that the inclusion of proportions enhances the informativeness of our findings (*Maxwell, 2010*). Finally, we note that we took a reflexive standpoint on our data, critically considering them within the broader context of the academic and policy literature and our professional knowledge and experience in this domain.

### Acknowledgements

We thank Kendra Levasseur for her excellent research assistance on this project.

**Sarah E Ali-Khan** is in the Centre for Intellectual Property Policy, Faculty of Law, McGill University, Montreal, Canada

(iD) https://orcid.org/0000-0002-9453-5086

**Liam W Harris** is in the Centre for Intellectual Property Policy, Faculty of Law, McGill University, Montreal, Canada

**E Richard Gold** is in the Centre for Intellectual Property Policy, Faculty of Law, and the Department of Human Genetics, McGill University, Montreal, Canada

richard.gold2@mcgill.ca

(iD) https://orcid.org/0000-0002-3789-9238

*Author contributions:* Sarah E Ali-Khan, Conceptualization, Data curation, Formal analysis, Investigation, Methodology, Writing—original draft, Project administration; Liam W Harris, Investigation, Writing—original draft; E Richard Gold, Conceptualization, Formal analysis, Supervision, Project administration, Writing—review and editing

*Competing interests:* E Richard Gold: This study was funded under the PACEOMICS project, supported by Genome Canada, Genome Quebec, Genome Alberta and the Canadian Institutes for Health Research, and by the Montreal Neurological Institute and Hospital (the MNI), which is part of McGill University, with which the authors are affiliated. The MNI identified the need for this study and approached one of the authors (ERG) to conceive, design and actualize the research. The MNI did not have access to the study data and played no further role in the study other than to supply a list of staff members. The other authors declare that no competing interests exist.

*Ethics:* Human subjects: This study was approved by the Research Ethics Board of McGill University. All interviewees provided written informed consent.

## Funding

| Funder | Grant reference number | Author |
| --- | --- | --- |
| Genome Canada | PACEOMICS | E Richard Gold |
| Canadian Institutes of Health Research | PACEOMICS | E Richard Gold |
| Genome Quebec | PACEOMICS | E Richard Gold |
| Montreal Neurological Institute | | E Richard Gold |
| Genome Alberta | PACEOMICS | E Richard Gold |

The funders had no role in study design, data collection and interpretation, or the decision to submit the work for publication.

## Additional files

### Supplementary files

• Supplementary file 1. The aggregated response data in this file shows the number of interviewees who mentioned each of the analytical categories developed by the authors through thematic analysis of qualitative interview data.

DOI: https://doi.org/10.7554/eLife.29319.003

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
