## [Decision Letter]

Thank you for submitting your article "Motivating Participation in Open Science by Examining Researcher Incentives" to *eLife* for consideration as a Feature Article. Your article has been reviewed by three peer reviewers, and the evaluation has been overseen by the *eLife* Features Editor (Peter Rodgers).

The reviewers and the Features Editor have agreed on the revisions that are needed to make the article suitable for publication in *eLife* (see list below).

The following individuals involved in review of your submission have agreed to reveal their identity: Cameron Neylon (Reviewer #2); Juan Pablo Alperin (Reviewer #3).

Summary:

This is a valuable contribution to the discussion on the implementation of Open Science approaches reporting on an important and large-scale experiment in this space. The study tackles an important issue facing the research community today: while most researchers agree with the principles of Open Science, there continue to be barriers to adopting OS practices that are deeply rooted in the academic culture. Exploring and understanding the motivations for adopting or resisting OS practices is therefore an important part of making changes. Neuro's OS framework does appear to offer an interesting case study to explore these issues. The methods are presented well, and I have no concerns about the way the data collection and data analysis are described. The choice of semi-structured interviews is justified convincingly.

Essential revisions:

1) The authors should discuss all ten of the themes they identify. There is substantial value to my mind in reporting the full set, as this will allow greater integration with the other literature in this space.

2) The issue of researchers' trust: The authors mention the role of incentives in publishing, and the significance of career outcomes for researchers. Publishing novel research can make researcher careers, while being 'scooped' may well break their careers. The concerns of scientists participating in open science initiatives often relate to the lack of trust in the system to safeguard their priority in publishing novel findings. I feel that the authors of this paper have somewhat ignored the issue of trust/distrust in science.

3) The authors need to do more to cite existing literature in the field and to place their own work in the context of previous work. Here are some specific examples:

3a) One very recent study which is conspicuously absent from the reference list, and which may inform the analysis, is: Open Research Data Taskforce (2017) Research Data Infrastructures in the UK: www.universitiesuk.ac.uk/policy-and-analysis/research-policy/open-science/Documents/ORDTF%20report%20nr%201%20final%2030%2006%202017.pdf

This is a comprehensive review of Open Science (preservation and sharing of data, the responsibilities of researchers, institutions and funders) in various countries, including Canada. Very helpfully, the report summarizes the problems the policy causes for researchers/scientists (though these were not based on interviews with scientists). Perhaps it would be useful for the authors to compare their findings with what the authors of the UK report found. Some points made here resonate with the UK study, but others stand in contrast to it. Is this because of the specific context of the country, or perhaps the context of this particular institution that appears to be head of the game in the implementation of Open Science policies?

3b) I think the paper could be substantially strengthened and made more generally applicable through reference to existing frameworks and similar efforts. I am thinking particularly of the work of Fecher and Friesikie (a number of articles including the important "Open Science: One term, five schools of thought" but also recent work on barriers to data sharing in Germany). Christine Borgman's work on data sharing (Big Data, Little Data…) and more broadly dissections of incentives (European Commission Expert Group report on role of research evaluation in progress to Open Science, Metric Tide report) and potentially the literature around choices to public Open Access as well could be valuable. At the moment I feel that the article reinforces what we have seen in other cases, which is valuable. I think it would be stronger and more valuable if these issues were developed in those broader contexts.

3c) (From Cameron Neylon). Something from my own recent work that I feel would be worth exploring is the extent to which group level dynamics help or hinder adoption. What is unique about the effort at the Neuro is the way it operates at a departmental level. Does being part of a group shift the dynamics of issues? Does identity of being in the department vs identity of being part of external communities contribute to engagement, or hinder it?

Note: This may be out of scope for this article, but I feel it could offer a productive line of inquiry that builds on the important and unique aspects of this program.

3d) The article would benefit from a more thorough literature review that looks at barriers to the adoption of open practices. While some aspects of Open Science are not as well studied, Open Access (OA) has been written about extensively in the literature, including studies about opinions and attitudes, and some explicitly about barriers to change. There have also been efforts to document barriers to adoption of Open Data. A few references are below; a number of publishers have also conducted surveys of researchers' awareness of and attitudes towards both open access and open data.

However, more important from my perspective would be to relate this other work to what has already been written in the Discussion section. For example, researcher's misgivings based on a lack of awareness and understanding of openness has been previously documented. Similarly, the endless studies on the OA Citation Advantage speaks to the author's point that the OS community has attempted to find evidence of the benefits to researchers.

Björk, B.-C. (2004). Open access to scientific publications-an analysis of the barriers to change? Retrieved from https://helda.helsinki.fi/handle/10227/647

Harley, D. (2013). Scholarly communication: Cultural contexts, evolving models. Science, 342(6154), 80-82.

Harley, D., Acord, S. K., Earl-Novell, S., Lawrence, S., & King, C. J. (2010). Assessing the Future Landscape of Scholarly Communication: An Exploration of Faculty Values and Needs in Seven Disciplines. Center for Studies in Higher Education. Retrieved from http://escholarship.org/uc/item/15x7385g

Peekhaus, W., & Proferes, N. (2015). How library and information science faculty perceive and engage with open access. Journal of Information Science, 41(5), 640-661. https://doi.org/10.1177/0165551515587855

Peekhaus, W., & Proferes, N. (2016). An examination of North American Library and Information Studies faculty perceptions of and experience with open-access scholarly publishing. Library & Information Science Research, 38(1), 18-29. https://doi.org/10.1016/j.lisr.2016.01.003

Xia, J. (2010). A longitudinal study of scholars attitudes and behaviors toward open-access journal publishing. Journal of the American Society for Information Science & Technology, 61(3), 615- 624. https://doi.org/10.1002/asi.21283

---

## [Author Response]

Essential revisions:1) The authors should discuss all ten of the themes they identify. There is substantial value to my mind in reporting the full set, as this will allow greater integration with the other literature in this space.

We have added data on all themes identified to the manuscript. These number nine not ten, as we combined some of them (notably on intellectual property and industry partners) to improve the flow of the text. Please see: Results section; Themes: “Current state of sharing”, “Attribution and publication”, “Infrastructure and resources” and “Research ethics”.

We also lightly edited parts of the existing themes to add to their comprehensiveness. Please see: “Open Science Needs to be Stakeholder Driven”, “Researchers: Motivated by ethics, but also career advancement” and “On patients”.

2) The issue of researchers' trust: The authors mention the role of incentives in publishing, and the significance of career outcomes for researchers. Publishing novel research can make researcher careers, while being 'scooped' may well break their careers. The concerns of scientists participating in open science initiatives often relate to the lack of trust in the system to safeguard their priority in publishing novel findings. I feel that the authors of this paper have somewhat ignored the issue of trust/distrust in science.

We have added data that are relevant to the issues of researchers’ trust to the Results and Discussion.

Specifically, please see:

- “Open Science Needs to be Stakeholder”: we report on the need for stakeholder consultation in open science policy design and for such policy to respect researchers’ academic freedom

- “Current state of sharing”: we note researchers’ agreement that they should be allowed to publish before releasing data

- “Researchers: Motivated by ethics, but also career advancement”: we note researchers’ fear of ‘scooping’ and misuse, and of the MNI policy going beyond cultural norms and thus disadvantaging them.

- Discussion: we discuss the issue of trust and how the MNI is starting to address this.

We hope that these additions properly address your point about trust.

3) The authors need to do more to cite existing literature in the field and to place their own work in the context of previous work. Here are some specific examples:3a) One very recent study which is conspicuously absent from the reference list, and which may inform the analysis, is: Open Research Data Taskforce (2017) Research Data Infrastructures in the UK: www.universitiesuk.ac.uk/policy-and-analysis/research-policy/open-science/Documents/ORDTF%20report%20nr%201%20final%2030%2006%202017.pdfThis is a comprehensive review of Open Science (preservation and sharing of data, the responsibilities of researchers, institutions and funders) in various countries, including Canada. Very helpfully, the report summarizes the problems the policy causes for researchers/scientists (though these were not based on interviews with scientists). Perhaps it would be useful for the authors to compare their findings with what the authors of the UK report found. Some points made here resonate with the UK study, but others stand in contrast to it. Is this because of the specific context of the country, or perhaps the context of this particular institution that appears to be head of the game in the implementation of Open Science policies?

Thank you for these helpful references. We have added citations to this report throughout the Results and Discussion. We felt that the issues it notes were largely consistent with our findings, but we have highlighted one or our findings that diverges from Fecher, Friesike and Hebings’ 2015 work and from the recent Wouters and Haak 2017 CWTS, University Leiden and Elsevier study.

3b) I think the paper could be substantially strengthened and made more generally applicable through reference to existing frameworks and similar efforts. I am thinking particularly of the work of Fecher and Friesikie (a number of articles including the important "Open Science: One term, five schools of thought" but also recent work on barriers to data sharing in Germany). Christine Borgman's work on data sharing (Big Data, Little Data…) and more broadly dissections of incentives (European Commission Expert Group report on role of research evaluation in progress to Open Science, Metric Tide report) and potentially the literature around choices to public Open Access as well could be valuable. At the moment I feel that the article reinforces what we have seen in other cases, which is valuable. I think it would be stronger and more valuable if these issues were developed in those broader contexts.

As noted directly above, we have added reference to Fecher, Freisike and Hebing, 2015 throughout the Results and Discussion, and to European Commission report, Wilsdon et al., 2017, and Borgman, 2015, to the Discussion. We do feel that our results are largely consistent with those reported by others. In the discussion however, we highlight the unique context of the MNI’s open science initiative, and reflect on how this may influence some of our findings.

3c) (From Cameron Neylon). Something from my own recent work that I feel would be worth exploring is the extent to which group level dynamics help or hinder adoption. What is unique about the effort at the Neuro is the way it operates at a departmental level. Does being part of a group shift the dynamics of issues? Does identity of being in the department vs identity of being part of external communities contribute to engagement, or hinder it?Note: This may be out of scope for this article, but I feel it could offer a productive line of inquiry that builds on the important and unique aspects of this program.

Thank you for this comment. Yes, we attempted to address this in the Discussion.

3d) The article would benefit from a more thorough literature review that looks at barriers to the adoption of open practices. While some aspects of Open Science are not as well studied, Open Access (OA) has been written about extensively in the literature, including studies about opinions and attitudes, and some explicitly about barriers to change. There have also been efforts to document barriers to adoption of Open Data. A few references are below; a number of publishers have also conducted surveys of researchers' awareness of and attitudes towards both open access and open data.However, more important from my perspective would be to relate this other work to what has already been written in the Discussion section. For example, researcher's misgivings based on a lack of awareness and understanding of openness has been previously documented. Similarly, the endless studies on the OA Citation Advantage speaks to the author's point that the OS community has attempted to find evidence of the benefits to researchers.Björk, B.-C. (2004). Open access to scientific publications-an analysis of the barriers to change? Retrieved from https://helda.helsinki.fi/handle/10227/647Harley, D. (2013). Scholarly communication: Cultural contexts, evolving models. Science, 342(6154), 80-82.Harley, D., Acord, S. K., Earl-Novell, S., Lawrence, S., & King, C. J. (2010). Assessing the Future Landscape of Scholarly Communication: An Exploration of Faculty Values and Needs in Seven Disciplines. Center for Studies in Higher Education. Retrieved from http://escholarship.org/uc/item/15x7385gPeekhaus, W., & Proferes, N. (2015). How library and information science faculty perceive and engage with open access. Journal of Information Science, 41(5), 640-661. https://doi.org/10.1177/0165551515587855Peekhaus, W., & Proferes, N. (2016). An examination of North American Library and Information Studies faculty perceptions of and experience with open-access scholarly publishing. Library & Information Science Research, 38(1), 18-29. https://doi.org/10.1016/j.lisr.2016.01.003Xia, J. (2010). A longitudinal study of scholars attitudes and behaviors toward open-access journal publishing. Journal of the American Society for Information Science & Technology, 61(3), 615- 624. https://doi.org/10.1002/asi.21283

Thank you for these references. We have added reference to several of them.